# Green Space Quality Analysis Using Machine Learning Approaches

Jaloliddin Rustamov [1,*] , Zahiriddin Rustamov [2] and Nazar Zaki [3,*]

1 Department of Genetics and Genomics, College of Medicine and Health Sciences, United Arab Emirates University, Al Ain 15551, United Arab Emirates
2 Faculty of Computer Science & Information Technology, University of Malaya, Kuala Lumpur 50603, Malaysia
3 Department of Computer Science and Software Engineering, College of Information Technology, United Arab Emirates University, Al Ain 15551, United Arab Emirates
* Correspondence: jaloliddin.rus@gmail.com (J.R.); nzaki@uaeu.ac.ae (N.Z.)

**Abstract:** Green space is any green infrastructure consisting of vegetation. Green space is linked with improving mental and physical health, providing opportunities for social interactions and physical activities, and aiding the environment. The quality of green space refers to the condition of the green space. Past machine learning-based studies have emphasized that littering, lack of maintenance, and dirtiness negatively impact the perceived quality of green space. These methods assess green spaces and their qualities without considering the human perception of green spaces. Domain-based methods, on the other hand, are labour-intensive, time-consuming, and challenging to apply to large-scale areas. This research proposes to build, evaluate, and deploy a machine learning methodology for assessing the quality of green space at a human-perception level using transfer learning on pre-trained models. The results indicated that the developed models achieved high scores across six performance metrics: accuracy, precision, recall, F1-score, Cohen's Kappa, and Average ROC-AUC. Moreover, the models were evaluated for their file size and inference time to ensure practical implementation and usage. The research also implemented Grad-CAM as means of evaluating the learning performance of the models using heat maps. The best-performing model, ResNet50, achieved 98.98% accuracy, 98.98% precision, 98.98% recall, 99.00% F1-score, a Cohen's Kappa score of 0.98, and an Average ROC-AUC of 1.00. The ResNet50 model has a relatively moderate file size and was the second quickest to predict. Grad-CAM visualizations show that ResNet50 can precisely identify areas most important for its learning. Finally, the ResNet50 model was deployed on the Streamlit cloud-based platform as an interactive web application.

**Keywords:** green space; quality; machine learning; image classification; ResNet50

## 1. Introduction

Green space is defined as green infrastructure containing vegetated areas, including grass, lawn, flowers, trees, parks, gardens, and forests [1,2]. Green space plays a vital role in aspects of daily life. Firstly, green space is associated with humans' improved physical and mental health. Secondly, it provides opportunities for social interactions and encourages physical activities such as walking. Thirdly, green space helps the environment by improving air quality, increasing urban biodiversity, and assisting in microclimate regulation [3–7]. Thus, the monitoring, analysis, and evaluation of the quality of green spaces are critical.

Green space at the human perception level represents how people perceive and experience green space [8]. The survey conducted by [9] found that the visual appearance of green space was the most crucial aspect of green space users' satisfaction. Paper [10] demonstrated that green space quality tends to serve as a determinant of people's desire to utilize green space and the benefits they derive from doing so, which assesses the quality of green space seen at the human perception level of utmost importance.

The quality of green space refers to the condition of the green space. It measures how well the site is maintained and the amenities it provides to make it safe, appealing, and inviting to visitors [11]. According to the survey by [12], cleanliness, maintenance, quietness, and safety were the essential qualities of green space. Other studies have confirmed that cleanliness, maintenance, and safety are the essential qualities of green space [13–16]. There is convincing evidence that poor, or a lack of, maintenance, such as littering, vandalism and dirtiness, may negatively affect green space usage. Appearance, concerns about safety, and the social setting of green space are crucial to its desirability to users [11]. According to [17], the maintenance of green space is costly and requires hard labour. The maintenance tasks of green space include removing litter, watering the trees and plants, raking leaves, removing old and dying trees and plants, and planting new ones. Central Park in New York City spent approximately 22 million US dollars on staff, maintenance, and other operations in 2021 [18]. With the advancements in Machine Learning (ML), recent studies proposed building ML models to analyze and assess green space, which could be used to automate some of the maintenance or monitoring tasks—saving time and reducing costs and labour required as a result [19].

One of the most prevalent approaches for evaluating green space is based on remote sensing data such as the Normalized Difference Vegetation Index (NDVI), which assesses the quantity of green space on satellite images [5,20]. There would be flaws if this method were employed to assess the quality of green space. Firstly, remote sensing data are still incapable of identifying small changes in quality-related variables [21]. Furthermore, satellite images are low resolution and provide a two-dimensional view of objects' top surface, which "may significantly differ from surrounding green space at the eye level" [2,20,22]. Secondly, as [23] suggested, green space quality is more important than green space quantity to people's health. Questionnaires and observations are the other way to assess the quality of green space. Both approaches are labour-intensive, time-consuming, and challenging to apply to a vast study area [23]. Hence, this research aims to assess the quality of green space using a machine learning approach on green space images captured at eye level by building an image classification model and evaluating its performance to ensure correct classification capability. Furthermore, this research aims to ensure that the proposed methodology can be helpful to the public by deploying it as an interactive web application.

Despite the relationship between green space and health being discussed in this paper, it will only focus on building, evaluating, and deploying a methodology encompassing image classification models to assess the quality of green space in Kuala Lumpur, Malaysia.

### 1.1. Research Motivation

As we discussed earlier, the quality of green space is an essential determinant of its usage. Upon conducting a literature review to explore the existing methods to assess the quality of green space, we discovered that most of the literature focuses on assessing the quantity of green space visible within manually collected or street view images using machine learning. Given the lack of studies conducted to assess the quality of green space, we proposed conducting a study to assess the quality of green space using machine learning, specifically, an image classification approach.

### 1.2. Research Contributions

The main aims of this study are as follows:

- Construct a green space image dataset comprising three classes: contaminated, healthy, and dried.
- Perform image augmentation techniques to balance the green space image dataset.
- Develop nine image classification models using transfer learning to classify green space images.
- Evaluate the learning performance of developed models using Grad-CAM.
- Deploy the best-performing image classification model on the Streamlit cloud-based framework for public use.

### 1.3. Section Organization

The organization of the remaining portion of the paper is as follows: Section 2 covers literature related to green spaces and the traditional and machine learning approaches used to evaluate their quality, Section 3 presents the proposed methodology and accompanying discussions, Section 4 displays the results and discussions of the experiments, and finally, the paper concludes in Section 5.

## 2. Literature Review

Green spaces are described as the ground that is partially or entirely covered with some form of vegetation [24]. Green space has the potential to contribute favourably to several of the most critical urban goals, including social inclusion, health, sustainability, and urban revitalization. Green space plays a crucial part in the day-to-day lives of residents. Environmental improvement through maintaining and expanding green space systems makes places more aesthetically pleasing and hospitable. It contributes to biodiversity preservation, promoting inward investment and increasing land values. Green spaces can catalyze broader communal and economic effects in a way that other neighbourhood facilities or structures cannot achieve. The fact that parks give free, open, non-discriminatory access 24 h a day, seven days a week and are apparent indicators of the quality of a neighbourhood were cited as significant aspects of their unique function [25].

Plants can indirectly influence the microbiome of the environment to which humans are exposed. Humans derive essential health advantages from the gut microbiome by controlling immunological balance and preventing chronic inflammation [24,26]. There is compelling evidence that higher accessibility to green space was linked to lower exposure to air pollution. Decreased exposure to air pollution and high accessibility to green space has proven to affect the cognitive development of children [24,27] positively. Increased exposure to green space has shown several physical health benefits, such as reducing the likelihood of cardiovascular and respiratory diseases in men, improved life expectancy, restoration of the brain's cognitive functionality, increased newborn babies' weight, lower risk of preterm birth, and higher self-reported health [24,27,28]. In addition, green space is proven to benefit mental wellbeing by improving mood and self-esteem, reducing stress levels, and reducing the risk of depression and anxiety [24,27–29].

Green space creates a communal area accessible to all segments of society. It can serve as the focal point of a community by providing numerous chances for social interaction, leisure, and recreational purposes. Few studies have shown the direct relationship between green space and health by improving social interaction [24,25,30]. Gardens can be a place for people to interact with each other; parks facilitate physical activities and leisure, and forests can be used for recreational activities. Isolated individuals are typically less healthy and more susceptible to stress, depression, and cardiovascular disease [30].

According to [31], urban green spaces help the conservation of biodiversity, making them an essential component of environmental protection. They provide various beneficial ecosystem services, including moderating climate extremes and reducing pollution by carbon sequestration. Carbon sequestration is the process of transferring and storing carbon dioxide in carbon pools [32]. Furthermore, green spaces help the environment by filtering dust, dirt, soot, smoke and liquid droplets, protecting the ozone layer against the ultraviolet (UV) radiation, lowering the impact of fierce winds, preventing erosion and pollution, and having beneficial effects on the natural water cycle, constraining storm-water runoff, protecting rivers from pollution, and reducing noise. They are necessary for the long-term sustainability of the environment [25,31]. According to [33], green space has been proven to help reduce the temperature in parks to lower than that in non-green areas.

### 2.1. Green Space Quality Analysis Using Traditional Methods

In a study conducted by [34], the authors proposed to find the relationship between the quality of green space and the frequency and duration of self-reported physical activities and self-reported stress, mental and physical health. The study surveyed 420 people

in Aydin, Turkey's parks and urban greenways. The study considered several quality factors: the distance to green space, aesthetics, cleanliness, size, maintenance, shaded areas, lighting, and openness/visibility. The results of the survey were analyzed using multivariate linear regression. The first finding of this study showed that distance to green space was negatively correlated to the number of users and frequency of physical activities. Secondly, cleanliness and maintenance were positively associated with the frequency of physical activities in green spaces. Finally, the size of green space was associated with less stress, and open/visible green space was associated with better physical health. In the research by [35], the authors conducted a study to evaluate the quality of Bucharest's green spaces. The study surveyed 51 citizens about the five parks under investigation using questionnaires with ten questions in the city of Bucharest. The study evaluated the parks based on the following criterion: green space placement (pollution, distance from home, and territory expansion), green space use (existence of recreational facilities and working places), environment (presence of water sources, shades, and space for pets) and biodiversity (diversity of vegetation and bird species). The weight of each criterion was determined using the Analytic Hierarchy Process (AHP) method. Each criterion was assessed using a five-point Likert scale. The study's initial findings showed that the five parks evaluated in this study were polluted to some extent. Two parks were in the city centre, making them reachable. However, only three of the parks had the freedom to expand their territory. Subsequently, all the parks offered recreational facilities and working places within their territory. Furthermore, a water source and special pet space were available in all the parks. However, only two parks offered areas covered with trees (shaded areas). Finally, the species and habitats were not diverse in the assessed parks. The author conducted a study to evaluate the aesthetic quality of green space while considering a human multi-sensory perspective and presented a systematic way to capture green space images to estimate near-view scenic beauty. The study surveyed a random selection of 178 people by using photo panels and a questionnaire at different sites in the Hangzhou Flower Garden in Hangzhou City, China. The study considered the following criteria for assessing the aesthetic quality of green space: visual, auditory, tactile, and olfactory. The study took the quantitative holistic approach to assess the landscape aesthetic. The authors captured 420 photos of bonsais and flowers and grouped them into panels with 12 photos per panel. The garden visitors were shown panels at the sites where the photos were taken. Then, they were asked to rank the 12 photos on a scale of 1–10 on their ability to represent the site. Subsequently, the five best-representing photos were selected from each panel and randomly assigned to 14 panels with 12 photos per panel. The 14 photo panels were randomly shown to the respondents, who were asked to rank ten photos in each panel from best to worst in terms of visual quality. The study's findings demonstrated that scenic beauty could offer an environment for relaxation for garden users. In addition, the findings showed that green space offered various aesthetic qualities, such as: auditory, olfactory, tactile, and visual. Firstly, green space provided auditory diversity, which is not offered in urban environments. Secondly, green space offered natural fragrances, which respondents admired. Thirdly, respondents appreciated some elements of the green space more when they touched them due to their tactile qualities. Finally, a defined way of taking, selecting, and presenting photos in a panel could eliminate bias and professional inability. In a recent study conducted by [12] to analyze the association between different features of green space and perception of green space qualities using the results of a survey and GIS-based spatial metrics. The study surveyed respondents in the form of an online and on-site questionnaire to assess the perceived importance of green space qualities in Brussels between 2015 and 2016. The survey yielded 371 responses, of which 349 were complete and valid. The green space types studied in this paper were 19th-century formal green spaces, public areas of housing projects, gardens, and spaces for community activities. The study considered nature and biodiversity, quietness, historical and cultural value, spaciousness, facilities, cleanliness and maintenance, and safety as quality factors under study. The survey results showed that cleanliness and maintenance, quietness, and safety are perceived as the most

important qualities of green space, followed by adequate facilities and spaciousness. The research by [36] examined neighborhood residents' perceptions of the quality and useful purposes of green spaces concerning neighborhood satisfaction and wellbeing. The study surveyed two neighborhoods (De Hoogte and Corpus-Noord) in Groningen, Netherlands, using a paper-mailed questionnaire in June 2014. Out of the 2750 questionnaires distributed, only 276 were returned, and 223 were completed. The survey results were analyzed using statistical, mediation, and linear regression methods. The 95% confidence interval was calculated using the Monte Carlo method. The quality factors studied in this paper were recreational facilities, amenities for a picnic, good natural features, the absence of litter, easy accessibility, and maintenance. The study results showed that residents with easy accessibility and usable green space were more content with their neighborhood. A study by [37] aimed to address the limitations of existing methods for assessing street greenery, such as questionnaires and field audits. The study aimed to use Google Street View (GSV) images to assess street greenery's eye-level quantity and quality. The study focused on the street greenery and evaluated the greenery, absence of litter, maintenance, and general condition by using a five-point scale. The data collection method was GSV images and field observation by a trained researcher. The study included a total of 240 streets in Hong Kong, China. The results indicated that the average quality of street greenery is 3.21 on a scale out of 5, indicating a relatively high quality. Furthermore, the findings showed that the quality of street greenery was linked to higher levels of physical activity in green spaces. The paper by [3] addresses the scarcity of existing multi-dimensional quality assessment tools for urban green spaces by developing and implementing the RECITAL tool. The study's objective was to assess the quality of green space and evaluate the reliability and internal consistency of the tool. The study focused on municipalities and urban areas. The quality factors assessed included surroundings, access, facilities, amenities, aesthetics and attractions, incivilities, potential usage, land covers, and animal and bird biodiversity. The study was conducted in Barcelona, Spain, where eight technicians conducted fieldwork, visiting between three and five green spaces per day, and completed a questionnaire for each space using the RECITAL tool. The study results showed that the tool was reliable, with an overall intraclass correlation coefficient (ICC) of 0.84, indicating a good reliability. The paper by [38] presented a study to create a tool to evaluate the quality of local green spaces known as "neighbourhood" green spaces. The study recognized that the current techniques for evaluating the quality of green spaces may not be suitable for smaller, local green spaces because these areas had different functions compared to the larger green spaces that people visit. The study's objective is to create a straightforward method to analyze and evaluate the quality of local green spaces, referred to as "neighbourhood" green spaces. The study focuses on neighborhood green spaces, and the quality factors assessed were appearance, maintenance, and the quality of various features. The data collection method used in the study was a survey conducted in Stoke-on-Trent, in the United Kingdom. The study was divided into three phases: phase 1 included four focus groups with 35 adults to gain opinions about local green space, phase 2 included a survey using a five-point scale on 635 adults to determine the appropriate weighting for various domain factors based on their relative significance, followed by testing for feasibility and reliability, and phase 3 included two researchers separately evaluating 28 local green spaces that met the established criteria for inclusion in the study. The study results showed that, according to survey participants, incivilities such as litter, dog waste, and vandalism were consistently deemed the most critical factors in determining the use of green spaces. The study by [39] aimed to analyze the relationship between physical activity and the access to high-quality urban green spaces. The study was conducted in Norwich, England and collected data from a questionnaire on self-reported physical activity levels of 4950 residents. The quality factors assessed included accessibility, maintenance, recreational facilities, amenity provision, signage and lighting, landscape, usage, and atmosphere. The study used multiple regression models to determine the relationship but found a lack of clear connections between leisure activities and green spaces. The paper by [40] aimed to investigate the relationship between the

quality of green spaces and prosocial behavior in children over time. The study focused on the quality of green spaces such as parks, playgrounds, and play spaces and assessed the quality factor of availability using a questionnaire and a Likert scale. The study's results showed that the quality of green spaces had a positive relationship with prosocial behavior in children. This research addressed the lack of conclusive evidence regarding the connection between neighborhood green spaces and prosocial behavior in children.

Table 1 summarises the literature review, outlining the type of green space studied, the quality factors assessed, and the method of assessment of the said quality factor.

**Table 1.** Summary of the literature review on Green Space Quality Analysis using traditional methods.

| Paper | Green Space Type | Quality Factors Assessed | Method of Assessment |
|---|---|---|---|
| [41] | Bonsais and flowers | Aesthetic quality | Quantitative holistic technique |
| [39] | Green space | Accessibility, maintenance, recreational facilities, amenity provision, signage and lighting, landscape, usage, and atmosphere. | Regression models |
| [12] | Green space, public areas | Nature and biodiversity, quietness, historical and cultural value, spaciousness, facilities, cleanliness and maintenance, safety. | Perception of green space users |
| [3] | Municipality and urban areas | Surroundings, access, facilities, amenities, aesthetics and attractions, incivilities, potential usage, land covers, animal biodiversity and birds' biodiversity. | Five-point Likert scale |
| [36] | Neighborhood | Recreation facilities, amenities for a picnic, natural features, absence of incivilities, accessibility, maintenance. | Statistical analyses |
| [38] | Neighborhood | Appearance, maintenance, and the presence of quality of various features | - |
| [35] | Park | Green space placement, green space use, environment, biodiversity | Five-point Likert scale |
| [40] | Park, playground | Availability | Likert scale |
| [34] | Park, urban greenway | Distance to green space, aesthetic, cleanliness, largeness, maintenance, shaded areas, lights, openness/visibility | Series of multivariate linear regression analyses |
| [37] | Street | Greenery, absence of litter, maintenance, general condition | Five-point Likert scale |

### 2.2. Green Space Quality Analysis Using Machine Learning

A study by [22] presented a novel methodology for classifying urban green spaces using a two-level system. The study aimed to improve upon existing methods for quantifying vegetation, such as NDVI, which lacked the resolution to detect smaller details like the presence of trash. The study's objective was to classify the land's health level and the presence of contamination in the green space. The quality factors used in the study were "Healthy", "Healthy Contaminated", "Dry", "Dry Contaminated" "Unhealthy", "Unhealthy Contaminated", "No Vegetation", and "No Vegetation Contaminated". The data for the study was collected using a DJI Phantom 4 drone, which captured 9901 aerial images from parks, university campuses, suburban neighbourhoods, and forested areas. The images were taken from ground level at 20–30 m. There were 9001 images used for training, and 901 images were used for testing. The study's authors designed their deep neural network consisting of a convolutional neural network for extracting features from the images and a multilayer perceptron acting as a classifier. The performance metrics used were accuracy, precision, recall, and an F1-score. The study results showed that the test accuracy, precision, and recall was 72%, while the F1-score was 71%. The research by [8]

aimed to develop a system for assessing the quality of urban street-level greenery using street-view images and deep learning. The key limitation with the existing methods, such as satellite and aerial imagery, was that they accurately quantified large-scale greenery but were weak at showing street-level greenery, including contours and features of ground plants. This research aimed to create a method for calculating and displaying the amount of visible greenery in urban areas at the street level. The researchers introduced the Panoramic View Green View Index (PVGVI) to do this. The research focused on parks and gardens, and assessed the quality factors like the visibility of greenery. The data source used in this research was the Cityscapes dataset, which contained recorded videos of streets from 50 different cities to benchmark the performance of their proposed method. Google Street View images were used to apply their proposed method to the study area. A total of 24,920 Google Street View images (1000 × 1000 pixels) were used in the study, which was conducted in Suita, Osaka, Japan. The algorithm used in this research was DeepLabV3+. The performance of the proposed method was evaluated using the mean intersection over union (mIoU), the root-mean-square error (RMSE), and the mean absolute error (MAE). The research results showed that the proposed method achieved an mIoU of 78.37%, an RMSE of 2.75%, and an MAE of 2.28%. The study by [23] presented a new machine-learning approach for evaluating the quality of street green space using street view images from Guangzhou, China. The study aimed to address the limitations of the current research methods for assessing green space quality, which was labour-intensive and time-consuming. Two thousand images were randomly selected for training purposes and were scored based on a 10-point scale of quality attributes by trained investigators. A random forest model was trained to automatically rate the images based on the proportion of 151 elements in the image segmentations. Two validation methods were used to evaluate the performance of the model, first the comparison of the automated scores with manually assessed images, and second by physical visits to residential neighbourhoods by three observers. The methods showed good consistency, whereby a Pearson correlation of more than 0.90 and an agreement percentage of over 85% was achieved. The study by [42] aimed to address the problem of urban planning practices overlooking the accessibility and visibility of street greenery. The study proposed to use Google Street View images to quantify street greenery and evaluated the discrepancy between visible greenery and street accessibility using space syntax. The study also evaluated the similarity between street greenery measurements, including visible and accessible greenery, and urban green cover obtained from satellite images. The study was conducted in Singapore using Google Street View images, a support vector machine (SVM) algorithm, and the scoring method of two urban planning experts to evaluate the results. Comparing the judgements of experts and the SVM showed a high level of accuracy with Cohen's Kappa coefficient values of 0.910 and 0.925. The study authored by [43], aimed to investigate the relationship between urban greenery and the time spent walking by pedestrians. The study specifically looked at the Green View Index (GVI), which measures the visibility of greenery from a specific position in neighbourhood streets. The study used a fully convolutional neural network for semantic segmentation (FCN-8s) to segment Google Street View (GSV) images, which were then used to calculate GVI. The model was trained on the Cityscapes dataset, with 22,973 images for training and 500 for validation. The study found that the model had a validation accuracy of 84.56%. The study by [44] examined the association between exposure to green and blue spaces in residential areas and geriatric depression in Beijing, China. The study aimed to address a gap in the knowledge about the relationship between the access to green and blue spaces and mental health in non-Western countries and the limitations of current methods for measuring exposure to these spaces. The study used deep learning techniques, specifically a fully convolutional neural network (FCN-8s), to segment street view images and compare the data to satellite imagery. The study trained the model using the ADE20K labelled image dataset and achieved a training accuracy of 81.4% and a test accuracy of 76.8%. In the study authored by [45], the problem addressed was that informal green spaces in urban areas, such as those used for recreation and forestry, are often small and not easily detected

through aerial surveys. As a result, these spaces were often overlooked by government and city planners during surveys and planning. The study's main objective was to test the feasibility of using machine learning to detect informal green spaces in Google Street View photos by applying the method in the study area of Ichikawa, Japan and Ho Chi Minh City, Vietnam. The study used 24,553 Green Space View panoramic images from Ichikawa, Japan and Ho Chi Minh, Vietnam, and 1000 manually labelled pictures were used to train the model. The DeepLabV3+ model was employed to classify and detect green areas in the GSV images, and the model's accuracy was 65%. The paper by [46] aimed to develop models that could predict the health of turf grass from aerial images as a solution to the limitations of visual examination, which may be subjective and influenced by personal biases. The study used 187 images, collected using a camera mounted on an unmanned aerial vehicle (UAV), and the quality factors evaluated were hue, texture, colour, leaf blade width, and uniformity. Three deep learning models were used for the prediction, AlexNet, GoogleNet, and Inception-V3. The performance was measured using accuracy and loss, and the results showed that Inception-V3 had the highest average accuracy of 73.35% and the lowest loss of 40.25%. The paper by [47] aimed to address the challenge of measuring the relationship between people's perceptions of the built environment and their health. The study focused on green space in the form of streets and the quality factors of nature quality, beauty, relaxation, and safety. The researchers used a dataset called PlacePulse 2.0, which contained 1.1 million images, and applied a SIAMESE CNN network. The results showed an average accuracy of 70.53%. The study by [2] focused on developing a natural language processing (NLP) application and text mining tool to evaluate the quality of urban green spaces. The data source used in the study contained 16,613 TripAdvisor reviews of St. Stephen's Green Park in Dublin, Ireland. The model used was a support vector machine (SVM). The performance of the model was measured using the area under the curve (AUC), precision, recall, and an F1-score, which showed a high performance with an AUC of 97.2%, a precision of 97.1%, a recall of 99.7%, and an F1-score of 98.3%.

Table 2 summarises the literature review, outlining the quality factors assessed by previous research, best-performing machine learning (ML) models, and their respective performance results.

**Table 2.** Summary of literature review on Green Space Quality Analysis using ML.

| Paper | Green Space Type | Quality Factors Assessed | ML Model | Results |
|---|---|---|---|---|
| [45] | Forest, park, cityscape | - | DeepLabV3+ | Accuracy: 65% |
| [2] | Park | - | SVM | AUC: 97.2%, precision: 97.1%, recall: 99.7%, F1-score: 98.3% |
| [8] | Park, garden | Visibility of greenery | DeepLabV3+ | mIoU: 78.37%, RMSE: 2.75%, MAE: 2.28% |
| [23] | Street | Accessibility, maintenance, variation, naturalness, colorfulness, clear arrangement, shelter, absence of litter, safety, general impression | Random Forest | Pearson's correlation coefficient: 0.90 |
| [42] | Street | Pedestrian and commuting accessibility, street greenery | SVM | Cohen's Kappa coefficient: 0.925 |
| [43] | Street | Visibility of greenery | FCN-8s | Accuracy: 84.56% |
| [44] | Street | - | FCN-8s | Accuracy: 76.8% |
| [47] | Street | Nature quality, beauty, relaxation, safety | CNN | Accuracy: 70.53% |
| [22] | Terrain | Health, contamination | Deep Neural Network | Accuracy: 72%, precision: 72%, recall: 72%, F1-score: 71% |
| [46] | Turfgrass | Hue, texture, color, leaf blade width, uniformity | Inception-V3 | Accuracy: 73.35%, loss: 40.25% |

Based on the summary of literature shown in Table 1, most past literature assessed the aesthetic qualities of green space using traditional methods. On the other hand, machine learning-based approaches assessed the quantity of visible green space, as presented in Table 2. Therefore, we proposed to assess green space's visual (aesthetic) quality using machine learning models in this research.

## 3. Methodology

The cross-industry standard process for data mining (CRISP-DM) is an iterative process model often depicted as a lifecycle model of projects [48,49]. This research adopted CRISP-DM as its methodology, as shown in Figure 1. First and foremost, the research problem was deduced from reviewing and analyzing past literature. Following that, the green space images were collected to be used as input to train the image classification models to address the problem determined. The performance of the trained models were evaluated using performance metrics such as accuracy, precision, recall and an F1-score. Finally, the best-performing model was be deployed as an interactive web application.

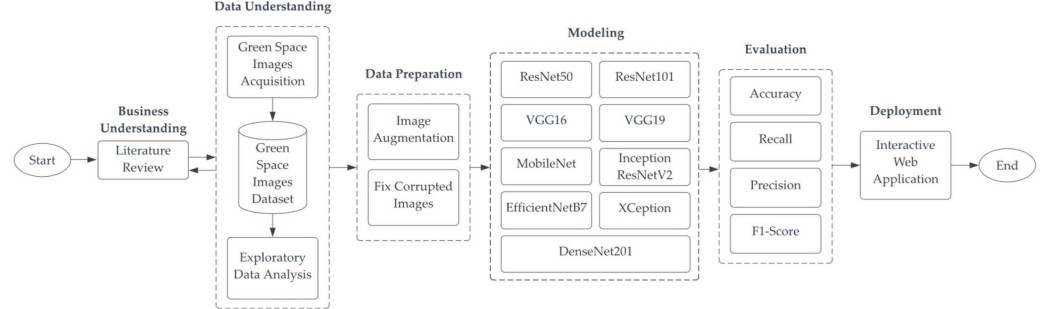

**Figure 1.** Flowchart of proposed methodology of this research.

Although CRISP-DM is a methodology commonly used with data mining projects, we used it for this research as it contains phases that coincide with the tasks we performed to carry out this research. Table 3 presents the tasks performed in each phase of the CRISP-DM methodology [50]. The particular tasks carried out in this research are discussed starting from Section 3.1.

**Table 3.** Description of each phase of the CRISP-DM Methodology.

| Phase | Description |
|---|---|
| Business Understanding | The project objectives and requirements are defined in this stage, and a preliminary plan is developed to address the business problem. This stage involves understanding the goals and objectives of the project and how the data mining solution will be used to solve the business problem. |
| Data Understanding | In this stage, data sources are identified, and data is collected, explored, and described. This stage involves getting familiar with the available data, its quality, and its limitations, and identifying any data issues that need to be addressed. |
| Data Preparation | Data is cleansed, transformed, and pre-processed in this stage to prepare for modeling. This stage involves selecting the relevant data, creating new variables, handling missing values, and addressing other data quality issues. |
| Modelling | In this stage, various modeling techniques are applied to the prepared data. This stage involves selecting and applying appropriate modeling techniques, evaluating model performance, and selecting the best model. |
| Evaluation | In this stage, the model is evaluated to determine whether it meets the business objectives. This stage involves assessing the model's performance, comparing it to other models, and assessing its generalizability and applicability. |
| Deployment | In this stage, the model is deployed into a production environment. This stage involves implementing the model, monitoring its performance, and making necessary adjustments. |

### 3.1. Description of Dataset

The green space images of Kuala Lumpur, Malaysia, were captured in July and August 2022. The location has a mean annual temperature of 25.8 °C and an annual precipitation of 2981 mm. The images were captured using the rear camera of the Apple iPhone XR. The specific configuration settings of the capture device and their purpose is shown in Table A1. Initially, the image dataset contained a total of 944 images of green space split into three different classes: Healthy, Dried, and Contaminated. The Healthy class contained images of green space in excellent condition that showed no signs of drying and no litter in its vicinity. The Dried class contained green space images that showed signs of dryness, and the Contaminated class contained green space images with litter in their vicinity. Figure 2 shows a sample of images from each class in the green space image dataset. The image dataset and codes can be found at: https://github.com/jaloliddin-rus/gsqualityanalysis (accessed on 15 February 2023).

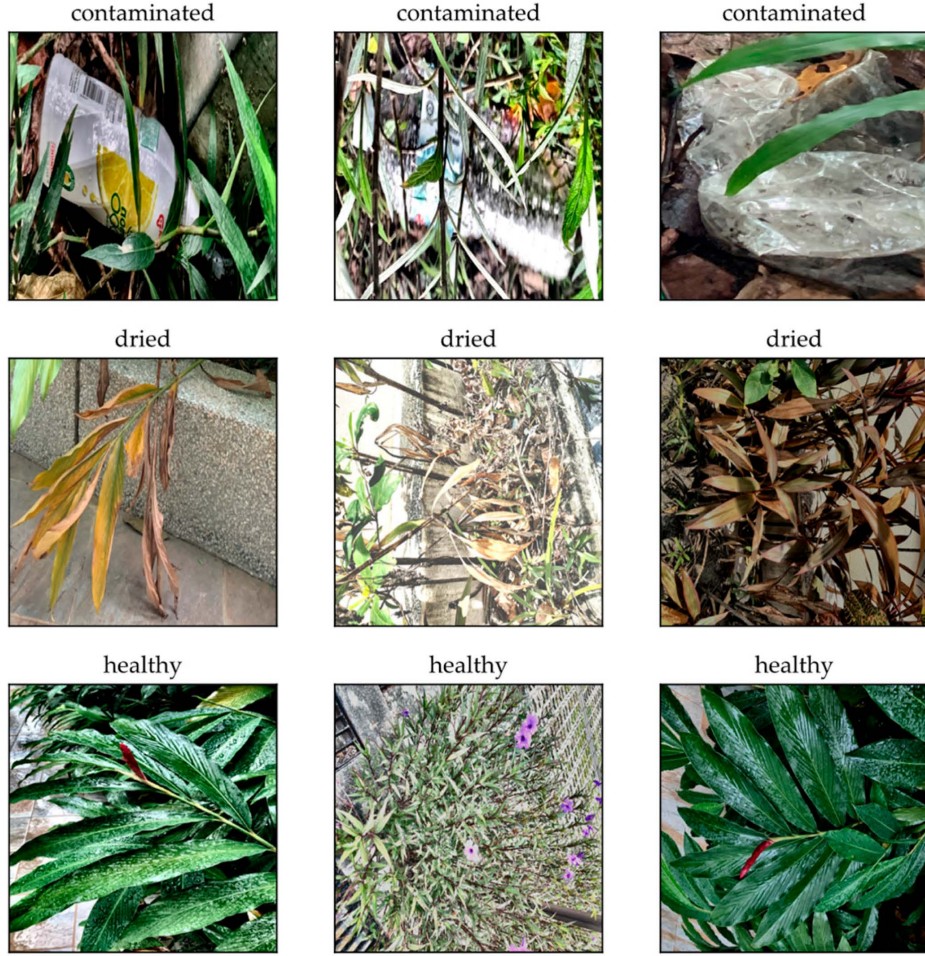

**Figure 2.** Sample images from the green space image dataset.

### 3.2. Exploratory Data Analysis

Figure 3 shows a bar chart of the initial number of images for each class in the green space image dataset. The exploratory data analysis indicated that the proportion of Dried and Contaminated images was less than the Healthy images. Due to the tropical rainforest conditions of Malaysia, it was more difficult to find green spaces that belonged to the Dried class than the Healthy class. Hence, the imbalance between classes.

Number of Images in Each Class



**Figure 3.** Bar chart of class balance pre-augmentation.

*3.3. Data Pre-Processing*

As shown in Figure 3, the Dried class had fewer images than the Healthy and Contaminated classes. Using the Albumentations library (version: 1.2.1) within the Jupyter Notebook, image augmentation was carried out to increase the number of images in the Dried and Contaminated classes to balance the number of images in each class to reduce the likelihood of bias and overfitting [51]. The images were augmented using transformation techniques presented in Table 4 with a 50% probability ($p = 0.5$) of applying the transformation.

**Table 4.** Image augmentation transformation applied.

| Transformation Technique | Parameter | Description |
|---|---|---|
| HorizontalFlip | $p = 0.5$ | Flips the supplied image horizontally. |
| RandomBrightnessContrast | $p = 0.5$ | Randomly alters the brightness and contrast of the image supplied. |
| RandomRotate90 | $p = 0.5$ | Randomly rotates the image supplied by 90 degrees zero or more times. |
| CLAHE | $p = 0.5$ | Applies Contrast Limited Adaptive Histogram Equalisation (CLAHE) to the image, enhancing an image's or video's visibility level [52]. |

Figure 4 presents a bar chart of the number of each green space image per class after performing image augmentation. As shown in Figure 4, the number of images per class is similar.

Number of Images in Each Class



**Figure 4.** Bar chart of class balance post-augmentation.

### 3.4. Model Training

The total number of images in the dataset post-image augmentation was 986. The image dataset was split into 70% training and 20% testing, and 10% validation data. The distribution of the green space images dataset is presented in Table 5. Before training the models, the green space images were resized to 224 × 224 pixels, the input size for all the models. The resizing is performed to lower the computational cost and does not significantly impact the models' accuracy [53].

**Table 5.** Dataset distribution for model training.

| Dataset Features | Value |
| --- | --- |
| Total observations | 986 |
| Total training data | 710 |
| Total validation data | 78 |
| Total testing data | 197 |
| Number of classes | 3 |

A transfer learning strategy was utilized for this research, and the following ImageNet weights were employed to train the models: ResNet50, ResNet101, DenseNet201, VGG-16, VGG-19, XCeption, MobileNet, InceptionResNetV2 and EfficientNetB7. Table 6 summarises the relevant hyperparameters set during model training. All algorithms and models were developed and implemented using the deep learning framework TensorFlow 2.9.0 with Keras API. The Categorical Cross-entropy loss function was used to train the model to measure its efficacy based on the probability of the truth. In addition, an Adam optimizer with a learning rate of 0.001 was utilized to minimize the loss function. We implemented an Early Stopping algorithm based on validation accuracy to ensure the model performance improved with each epoch. EarlyStopping would stop the training after four epochs of no improvement in validation accuracy. In addition, we utilized ReduceLROnPlateau, which decreased the learning rate if the validation accuracy was not improving after two epochs with no improvement. We used validation accuracy as a metric to monitor the two regularisation algorithms, as validation accuracy is calculated in each epoch from the dataset aside from the training dataset. The regularisation algorithms check the performance of the model after each epoch. Therefore, it is best to check the performance of a metric which is calculated in each epoch. A dense layer was built using SoftMax activation in the preceding layers to provide probability distributions for the Healthy, Dried, and Contaminated classes.

**Table 6.** Training hyperparameters.

| Parameter | Value |
| --- | --- |
| Data split | Training: 70%, Testing: 20%, Validation Split: 10% |
| Image size | 224 × 224 × 3 |
| Weight | ImageNet |
| Epochs | 50 |
| Batch size | 32 |
| Optimization algorithm | Adam (learning rate: 0.001) |
| Regularization algorithm | ReduceLROnPlateau (patience: 2, factor: 0.1), Early Stopping (patience: 4) |
| Loss function | Categorical Cross-entropy |

### 3.5. Transfer Learning

Transfer learning (TL) employs prior knowledge acquired from a source domain and task to enhance the performance of a model for a "similar" target domain and task, as contrasted to training just on the target domain and task from random initialization. While TL approaches are advantageous in many learning contexts, they excel when sufficient training data are unavailable in the target domain. However, equivalent source data may

be acquired from knowledge about the target domain. Therefore, TL allows for improved performance with fewer training data and trained models without retraining. For instance, it has been demonstrated that large datasets produce the best results, but large datasets take a long to obtain and are labour-intensive. Using prior knowledge from synthetic, augmented, or other captured training datasets, TL may enable equivalent performance with fewer captured training datasets [54].

### 3.6. Classification Models

CNN is mainly used for image classification as it can extract features by combining convolutional and pooling layers accompanied by fully connected layers and a SoftMax layer. The number of layers of the CNN dictates the ability to extract features, and the lower number of layers means a weaker ability to learn complex features [55,56].

Residual Network (ResNet) is a CNN-based model proposed by Kaiming He in 2015. ResNet50 and ResNet101 only differ by the number of layers they contain. As the name suggests, ResNet50 has 50 layers, whereas ResNet101 contains 101 layers. ResNet comprises convolutional, max pooling, average pooling, fully connected and SoftMax layers. ResNet model skips layers to ensure the gradient in the previous layer's performance does not reduce [57]. As CNN models tend to overfit when they go deeper in terms of layers, ResNet's architecture allows it to retain its performance regardless of the depth of the layers [58].

DenseNet201 contains 201 deep layers within its structure. As the name suggests, it features dense connections so that the $i$th layer comprises $i$ connections rather than a single connection from the preceding layer, as in a conventional feed-forward network. As a result, it eliminates the vanishing gradient problem, promotes feature propagation, enhances feature reuse, and considerably decreases the number of parameters [59].

The VGG-16 model achieved 92.77% accuracy with the ImageNet database consisting of fourteen million images and one thousand classes. VGG-16 is a CNN comprising 16 layers [60]. In addition, the VGG-16 has an input image size of 224 × 224 and consists of 13 convolutional layers and three fully connected layers, followed by max pooling and SoftMax [61]. VGG-19 is a modified version of VGG-16 which consists of 19 layers, including 16 convolutional layers, two fully connected layers, a single classification layer, and five max-pooling layers [62].

XCeption is short for extreme Inception [59]. It is an improvement over the traditional Inception model. Furthermore, it includes thirty-six convolutional layers and is the foundation for the feature extraction block. A residual network separates the convolutional layers and connects them [63]. The XCeption model implemented depth-wise separable convolution, which can significantly lower the cost of the convolution process [64].

MobileNet is an efficient CNN model for mobile and embedded systems, and it is a depthwise separable convolution model emphasizing reducing latency. Furthermore, it is a small network [59].

InceptionResNetV2 is a 164-layer CNN based on InceptionV3 and ResNet [65]. Residual connection is used because it eliminates degradation problems during deep structure and provides precise feature information such as texture, size, color, and placement. Multiple convolutions, pooling layers, and all feature maps that are concatenated into a single vector in the output section comprise the inception module. Typically, the module's filter sizes are 5 × 5, 3 × 3, and 1 × 1 for extracting local and global features from input images. ResNet is recognized for its shortcut connection, which efficiently summarizes the characteristics of the previous and subsequent layers [66].

EfficientNet is a pre-trained CNN-based model. It consists of eight variants, ranging from B0 to B7, with the higher the number, the greater the number of parameters. It comprises a minimal number of parameters to maintain accuracy. By consistently scaling and balancing the three parameters of neural network depth, width, and resolution, the EfficientNet model is typically more efficient and accurate than other CNN. It scales up to deep block layers instead of building a new CNN model from scratch [67].

Table 7 shows the comparison between the classification models used in this study in terms of the number of layers, a rough estimate of the time taken to predict (using an image size of 224 × 224 and a batch size of 1), model file size, presence of skip connections, and architectural complexity.

**Table 7.** Comparison between classification models used in this study.

| Model | Depth (no. of Layers) | Time Taken to Predict (ms) | Model File Size | Skip Connections | Architectural Complexity |
|---|---|---|---|---|---|
| MobileNet | 27 | 1–2 | Small | Yes | Simple |
| XCeption | 71 | 3–4 | Small | Yes | Simple |
| DenseNet201 | 201 | 10–11 | Small | Yes | Complex |
| ResNet50 | 50 | 2–3 | Medium | Yes | Simple |
| ResNet101 | 101 | 7–8 | Medium | Yes | Complex |
| InceptionResNetV2 | 164 | 8–9 | Medium | Yes | Complex |
| VGG-16 | 16 | 5–6 | Large | No | Simple |
| VGG-19 | 19 | 6–7 | Large | No | Simple |
| EfficientNetB7 | 813 | 25–30 | Large | Yes | Complex |

### 3.7. Evaluation Metrics

Evaluation metrics are used to assess the performance of the machine learning models. There are four categories in the evaluation metric to compare the actual result and predicted result: False Negative (*FN*), False Positive (*FP*), True Negative (*TN*), and True Positive (*TP*). Numerous evaluation metrics can be used to assess the performance of machine learning models [57]. This research uses the following four evaluation metrics: accuracy, precision, recall, and the F1-Score.

Accuracy measures all cases that correctly classified observations against total observations [57]. The accuracy of classification is determined using the following formula:

$$Accuracy = \frac{TP + TN}{TP + TN + FP + FN} \tag{1}$$

Precision measures the number of cases in the positive observations predicted correctly against the total observations [57].

$$Precision = \frac{TP}{TP + FP} \tag{2}$$

Recall measures the number of cases of correctly predicted positive observations against all observations in the actual class [57].

$$Recall = \frac{TP}{TP + FN} \tag{3}$$

The F1-Score shows the weighted average value of precision and recall providing values that cannot be classified by accuracy. It takes both false positives and false negatives into account [57].

$$F1\ score = \frac{2 * Precision * Recall}{Precision + Recall} \tag{4}$$

Cohen's Kappa is employed in classification to evaluate the level of agreement between observed and predicted classes. Cohen's Kappa score of 0 implies complete disagreement, whereas a score of 1 depicts total agreement [68].

$$K = 1 - \frac{1 - P_0}{1 - P_e} \tag{5}$$

The ROC curve shows how well a binary classifier can predict outcomes at different threshold values. The AUC measures the classifier's overall performance across all possible

thresholds, with a score of 1 indicating perfect accuracy and 0 indicating complete inaccuracy. The AUC determines the likelihood of the model performing better than random chance [69].

Gradient-weighted class activation mapping (Grad-CAM) provides an assessment and illustration. Grad-CAM utilizes the gradients of a given target concept to produce a rough localization map that accentuates significant areas for image classification [70].

### 3.8. Deployment

An interactive web application was developed to deploy the trained CNN model to assess the quality of green space. It was developed to facilitate the immediate usage of CNN models in practical applications. The interactive web application was developed using Python programming language and Streamlit syntax on PyCharm and deployed on Streamlit for public use. Streamlit is an open-source Python-based framework for developing web applications for machine learning and data science [71]. The interactive web application works on mobile phones and computers. The interactive online application enables the user to either capture an image using the mobile phone's camera or select an image from the device's storage. The interactive web application feeds the image to the CNN model, which then classifies the image. Accordingly, the user gets prompted with the classification result and suggests an action be taken based on the classification result.

### 3.9. Tools

The computer's hardware configuration employed to train the machine learning models is shown in Table 8.

**Table 8.** Hardware configuration.

| Component | Brand and Model |
| --- | --- |
| Operating System (OS) | Windows 10 Professional Edition (Version 21H2, Build 19044.1826) |
| Central Processing Unit (CPU) | AMD Ryzen 7 5800X |
| Graphical Processing Unit (GPU) | Nvidia GTX 1080 Ti 11GB GDDR5 |
| Random Access Memory (RAM) | 32GB @ 3800 MHz |

The models were developed by coding using the Python (version: 3.7) programming language on a Jupyter Notebook (version: 6.4.12) using the Anaconda environment (version: 2.0.3). Furthermore, the models were trained using the TensorFlow library (version 2.9.0), CUDA Toolkit (version 11.2) and CUDA Deep Neural Network (cuDNN) (version 8.1.0)., This research used Streamlit (version 1.11.1) and JetBrains PyCharm (version: 2022.1.3) to deploy the interactive web application. Mendeley (version: 1.19.8) was used as a reference manager to maintain the references used in this research.

## 4. Results and Discussion

This section gives the results of several experiments on the green space image dataset with nine transfer learning networks, namely, DenseNet201, EfficientNetB7, Inception-ResNetV2, MobileNet, ResNet50, ResNet101, VGG-16, VGG-19, and XCeption. First, we evaluated the performance of these models using the performance metrics discussed in Section 3.7. Following, we compared the inference time and file size of the models to assist us in choosing the overall best-performing model for deployment. Finally, we discussed the details of the deployment stage and the challenges and limitations we faced in this research.

All the models in this study were developed using the same hyperparameters as shown in Table 6. The accuracy metric results are presented in Figure 5, showing that EfficientNetB7 had the highest accuracy among all the models. However, the difference between EfficientNetB7, ResNet101, and VGG-19 was minimal. Therefore, we evaluated the models using other performance metrics mentioned in Section 3.7.

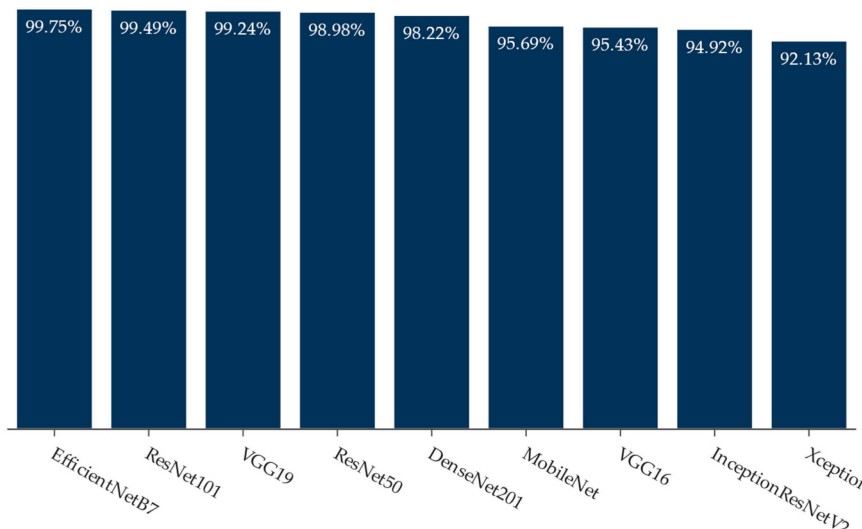

**Figure 5.** Bar chart of accuracy metric of models.

Table 9 shows the evaluation of all the models using the performance metrics. The results showed that all the models that were developed had performed exceptionally. Even the model with the lowest accuracy, XCeption, achieved an accuracy of 92.13%. Despite the small margin of difference between the results, EfficientNetB7 was the best-performing model as it achieved the highest values in the evaluation process. It achieved accuracy, recall, precision, and an F1-Score of 99.75%, Cohen's Kappa score of 1.00, and ROC-AUC of 1.00. As discussed in Section 3.7, Cohen's Kappa score and the ROC-AUC of 1.00 translated to the model being a perfect classifier. However, we did not conclude that EfficientNetB7 was the model to be deployed as we evaluated the models regarding inference time and file size. These were critical factors to consider before deployment, as they could affect the user experience and performance of the device.

**Table 9.** Performance metrics of models.

| Model | Accuracy (%) | Precision (%) | Recall (%) | F1-Score (%) | Kappa | ROC-AUC |
|---|---|---|---|---|---|---|
| XCeption | 92.13 | 92.13 | 92.13 | 92.12 | 0.88 | 0.98 |
| InceptionResNetV2 | 94.92 | 94.96 | 94.92 | 94.93 | 0.92 | 1.00 |
| VGG-16 | 95.43 | 95.43 | 95.43 | 95.43 | 0.93 | 0.99 |
| MobileNet | 95.69 | 95.81 | 95.69 | 95.70 | 0.94 | 0.99 |
| DenseNet201 | 98.22 | 98.24 | 98.22 | 98.23 | 0.97 | 1.00 |
| ResNet50 | 98.98 | 99.00 | 98.98 | 98.98 | 0.98 | 1.00 |
| ResNet101 | 99.49 | 99.49 | 99.49 | 99.49 | 0.99 | 1.00 |
| VGG-19 | 99.24 | 99.24 | 99.24 | 99.24 | 0.99 | 1.00 |
| EfficientNetB7 | 99.75 | 99.75 | 99.75 | 99.75 | 1.00 | 1.00 |

Next, we compared the file size of the models and accuracy as the file size was one of the crucial factors to consider when building and deploying models. The detailed comparison results are presented in Table A2 in the Appendix A.

These models were trained using a computer with a hardware configuration more powerful than the average smartphone. As a result, a larger model file size may reduce the inference time while using the model on a smartphone or a weaker device. If the model were to be integrated into a mobile application, the model would have to be shipped together with the application, which takes up storage space [72]. As shown in Figure 6, the file size of the MobileNet model was the smallest, as it was made to develop mobile-friendly models whereby file size and architectural complexity were of utmost concern. On the other hand, the file size of the EfficientNetB7 model was the largest, most likely due to its

architectural complexity and number of layers. As discussed in Section 4, EfficientNetB7 achieved the highest results when evaluated using performance metrics mentioned in Section 3.7. However, the file size of the EfficientNetB7 was not optimal for cloud-based platforms such as Streamlit due to limitations on computational resources. Next, we evaluated the inference time of the developed models.

As good as a model can perform during evaluation, the optimal inference time (time taken to perform the image classification) should be short. This is because taking a longer time has a negative impact on the user experience. Therefore, we evaluated each developed model's average time taken for image classification. Referring to Figure 7, MobileNet performed the image classification quickly, while VGG-16 and 19 took the longest. VGG-16 and 19 required more computational resources to make predictions due to their large number of parameters, rendering them less efficient than other models developed in this study [73–75].

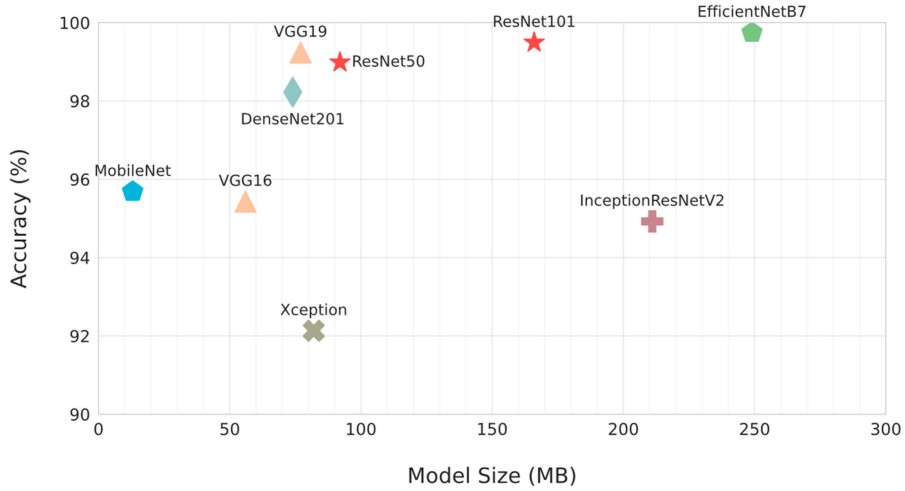

**Figure 6.** Comparison of accuracy against file size of the models.

On the other hand, MobileNet was the quickest due to its architectural design, which uses depthwise separable convolutions. MobileNet reduced the number of computations required in each layer by using depthwise separable layers, which resulted in faster predictions. Furthermore, it used fewer parameters than other CNN architectures like VGG and ResNet, reducing the computational cost of the model [76,77].

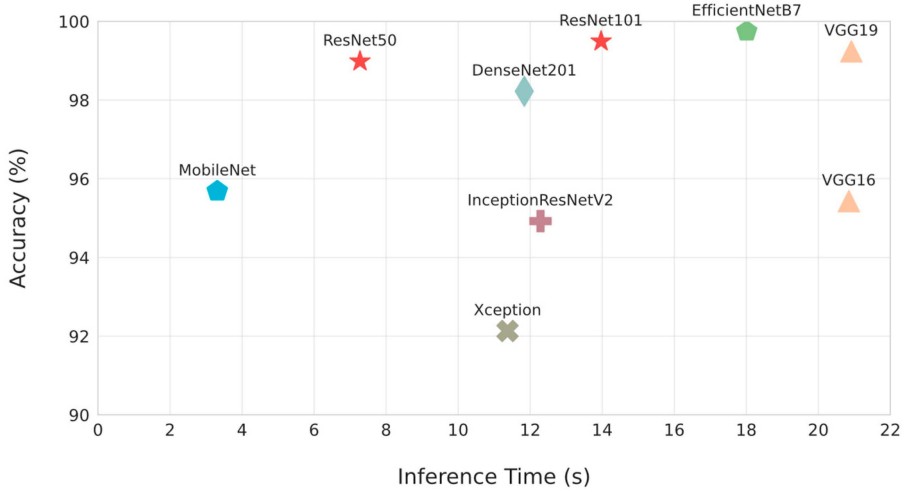

**Figure 7.** Comparison of accuracy against inference times of the models.

In order to understand how a machine learning model learns from different types of data, it is crucial to evaluate the effectiveness of its training. As a result, in our experiments, we showcased the learning performance of the developed models using Grad-CAM heat maps. The images selected from the dataset for Grad-CAM visualization are presented in Figure 8. Figures 9–11 show the learning performance of the models for each class, along with heat maps on the images. The areas which were considered most important were highlighted with a yellow color. These figures show that EfficientNetB7, ResNet50, and ResNet101 models could precisely identify and learn from areas unique to each class. Additional Grad-CAM visualizations are included in Appendix B.

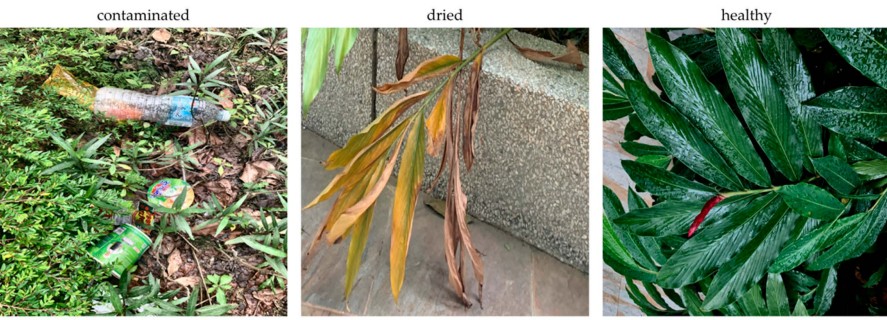

**Figure 8.** Images selected from the dataset for the Grad-CAM visualization.

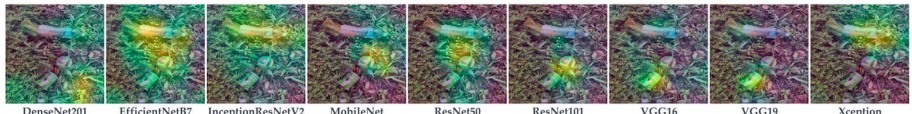

**Figure 9.** Grad-CAM visualization of classification of the Contaminated class.

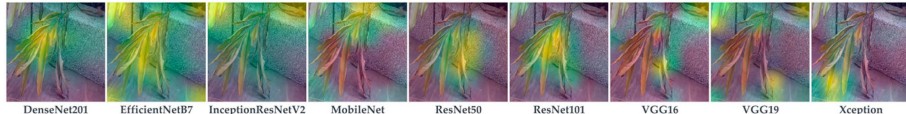

**Figure 10.** Grad-CAM visualization of classification of the Dried class.

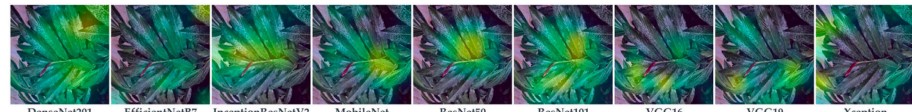

**Figure 11.** Grad-CAM visualization of classification of the Healthy class.

Based on our findings, we could observe that EfficientNetB7, ResNet101, and ResNet50 performed exceptionally well during our evaluation. However, the file size and inference time of EfficientNetB7 and ResNet101 were not optimal for the cloud-based interactive application. Therefore, we chose to deploy the ResNet50 model on the interactive web application as it performed exceptionally well during evaluation. It had a relatively moderate file size and was the second quickest to predict, with a 98.98% accuracy.

### 4.1. Deployment

The proposed model was deployed using the Streamlit framework as an interactive web application. The web application can be accessed using a computer or smartphone. Users can capture an image using the camera and upload or use an image in the device's storage. The step-by-step manual guide on operating the web application is shown in Figures A4 and A5 in Appendix C.

Figure 12 shows a partial screenshot of the interactive web application interface and classification result. The image of dried leaves captured in real-time through any

device with internet browser capabilities was uploaded to the web application, and the classification of the uploaded image was displayed as a result. Furthermore, the web application also showed the prediction breakdown, which showed the probability of each class.

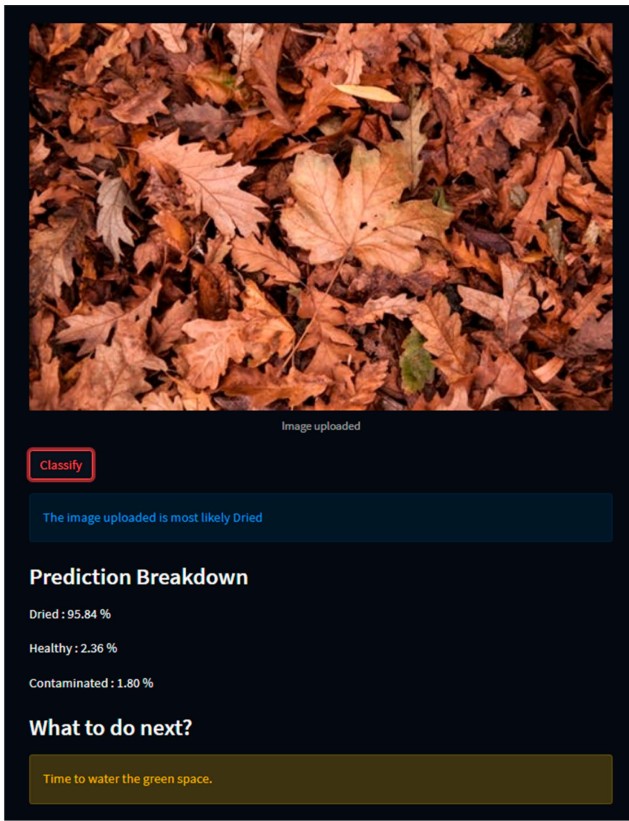

**Figure 12.** Screenshot of interactive web application.

### 4.2. Challenges and Limitations

Firstly, healthy green space, which has a similar color to dried green space, impacts the classification performance. For example, the sun shines light directly onto the healthy green space and the color of the healthy green space changes to yellowish green, a feature of dried green space. Secondly, image classification is performed on the whole image, regardless of the presence of green space, image-to-visible green space ratio, or similarity of green space with other objects. Assessing the quality of green space using object detection or image segmentation could be a better approach as it would allow the observation of these unnoticed features within an image.

### 5. Conclusions

In this research, an image classification model to assess the quality of green space was proposed, trained, evaluated, and implemented. For this objective, an image dataset of 944 images was collected. The dataset consists of images of Healthy, Dried, and Contaminated green spaces. For the data pre-processing, we performed image augmentation to increase the number of images to balance all the classes. Given the small size of our dataset, we employed transfer learning on the pre-trained models, including EfficientNetB7, ResNet50, ResNet101, MobileNet, VGG-16, VGG-19, XCeption, InceptionResNetV2 and DenseNet201. The results show that EfficientNetB7 achieved the highest result using six evaluation metrics implemented in this study. However, the difference in the performance among the top three models is very tiny. Since the model would be deployed on a cloud-based platform, selecting the best-performing model based on evaluation using performance metrics is

not ideal. Therefore, we evaluated the models by comparing their file size and inference time. Our findings show that the ResNet50 model is the most suitable to deploy as it has a moderate file size and the second fastest inference time with high accuracy of 98.98%.

For future work, we plan to develop a mobile application to assess the quality of green space, allowing people to capture images of green space and report the image of green space with the precise location to the respective authorities to take necessary action. Furthermore, given the limitations of image classification, we would like to develop an image segmentation model as it can provide a precise outline of the object within an image.

**Author Contributions:** Conceptualization, Z.R.; Data curation, J.R.; Funding acquisition, N.Z.; Methodology, J.R.; Software, Z.R.; Visualization, J.R.; Writing—original draft, J.R.; Writing—review and editing, Z.R. and N.Z. All authors have read and agreed to the published version of the manuscript.

**Funding:** This work is supported by the United Arab Emirates University through the UAEU-ZU Joint Research Grant G00003819 under the Emirates Center for Mobility Research.

**Institutional Review Board Statement:** Not applicable.

**Informed Consent Statement:** Not applicable.

**Data Availability Statement:** Not applicable.

**Acknowledgments:** The authors would like to thank the United Arab Emirates University for funding this work under the UAEU-ZU Joint Research Grant G00003819 through the Emirates Center for Mobility Research.

**Conflicts of Interest:** The authors declare no conflict of interest.

## Appendix A

The dimensions refer to the size of the image captured in pixels. The focal length is the distance that separates the lens's optical centre from the camera sensor; the F-stop determines how much light enters the sensor. At the same time, exposure time measures the duration of time that the camera's shutter remains open to allow light to enter and reach the camera's sensor. ISO refers to the sensitivity of the sensor to light, and exposure bias is the adjustment of the camera's exposure settings to over- or under-expose the image.

**Table A1.** Capture device configuration.

| Image Properties | Value |
| --- | --- |
| Dimensions | 3024 × 4032 pixels |
| Device Manufacturer & Model | Apple iPhone XR |
| Focal Length | 4.25 mm |
| F-Stop | f/1.8 |
| Exposure Time | 1/500 s |
| ISO | ISO 25 |
| Exposure Bias | 0 |
| Flash Status | No Flash |

**Table A2.** Average inference time and file size of the developed models.

| Model | Avg. Inference Time (s) | File Size (MB) |
| --- | --- | --- |
| DenseNet201 | 11.84 | 74 |
| EfficientNetB7 | 18.01 | 249 |
| InceptionResNetV2 | 12.29 | 211 |
| MobileNet | 3.31 | 13 |
| ResNet50 | 7.28 | 92 |
| ResNet101 | 13.97 | 166 |
| VGG-16 | 20.85 | 56 |
| VGG-19 | 20.92 | 77 |
| XCeption | 11.37 | 82 |

## Appendix B

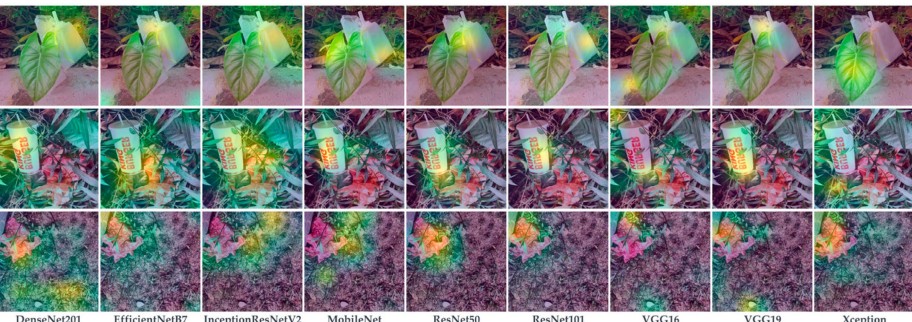

**Figure A1.** Additional Grad-CAM visualization of classification of the Contaminated class.

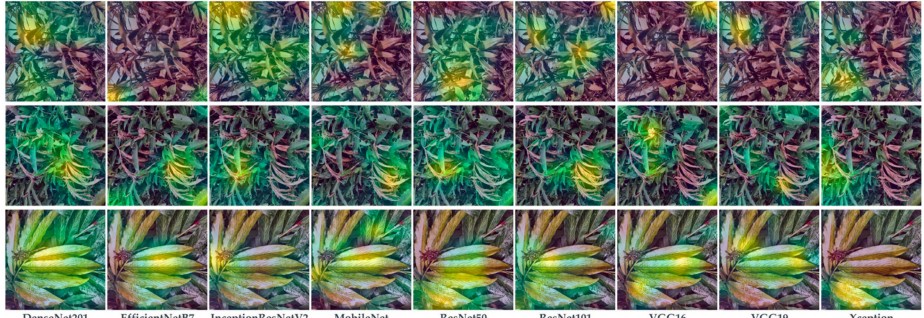

**Figure A2.** Additional Grad-CAM visualization of classification of the Dried class.

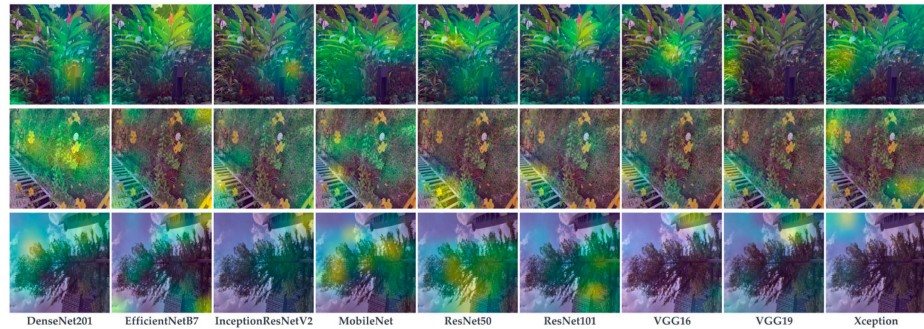

**Figure A3.** Additional Grad-CAM visualization of classification of the Healthy class.

## Appendix C

The web application can be found at the following link: https://greenspace.streamlit.app/ (accessed on 21 January 2023).

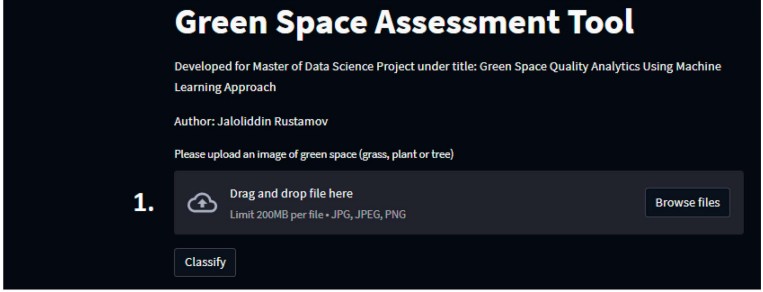

**Figure A4.** Screenshot of the upload module from interactive web application.

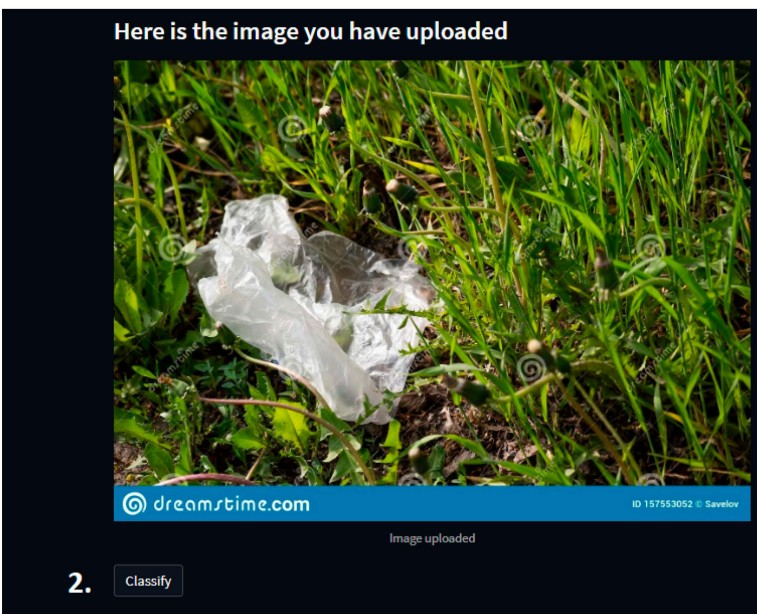

**Figure A5.** Screenshot of the prediction module from interactive web application.

1. Select an image from the device's storage or capture an image using the device's camera.
2. Press the *Classify* button and wait for the result.

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
