# Peer review of "Green Space Quality Analysis Using Machine Learning Approaches"

_sustainability, doi:10.3390/su15107782_

Round 1

Reviewer 1 Report

Article ID: sustainability-2255616

Article Title: Green Space Quality Analysis using Machine Learning Approaches

In this paper, this researcher proposed to build, evaluate, and deploy a machine learning methodology for assessing the quality of green space at a human-perception level using transfer learning on pre-trained models. The results indicated that the implemented models achieve high scores across four performance metrics: accuracy, precision, recall, and F1-score. Moreover, the models are evaluated  for their file size and inference time to ensure practical implementation and usage. The best-performing model, ResNet50, achieved 97.6% accuracy, 98% precision, 98% recall, and 98% F1-score.  The deployed interactive web application can be used on a computer and a smartphone.

Therefore, it is interesting and attractive. However, it should be major revised to enhance the quality, as follows:

1) In Section 1, authors should make three sub sections, motivation, contributions and organization of the paper

2) Literature review is not upto the mark. Pl include one table for comparison analysis 

3) A summary table should be provided for convenience for the readers in literature review section with comparison analysis of other approaches

4) Contributions of the research paper is limited, Pl at least three contributions should be there in any journal article

5) Overall work methodology.is not clear. Pl elaborate it clearly

6) Table 1 and 2 should be re-presented. Moreover, all the parameters should be explain clearly .

7) Section 4.2 should derive more

8)  All Figures should be enhanced at the resolution of 300 dpi.

9) Finally, the authors should double-check all formation, typos, and writing throughout the paper.

Author Response

Dear Sir/Madam,

Thank you very much for your time and comments. We have tried to address each comment to the best ability, and we made all the changes mentioned in your comments to the best ability. We have left our response to your kind comments in dark blue font colour below. Thank you for your kind suggestions.

Kind regards,

Jaloliddin Rustamov

  1. In Section 1, authors should make three sub sections, motivation, contributions and organization of the paper
    • We have added subsections for motivation, contributions, and organization of the paper in the Introduction section. (Line 83-105)
  2. Literature review is not up to the mark. Pl include one table for comparison analysis 
    • We have reviewed more papers utilizing traditional and machine-learning approaches for green space quality analysis. (Line 149-375)
  3. A summary table should be provided for convenience for the readers in literature review section with comparison analysis of other approaches
    • We have added a table after each literature review sub-section to summarise the literature review conducted. (Line 267; Line 369)
  4. Contributions of the research paper is limited, Pl at least three contributions should be there in any journal article
    • We have added the list of contributions in the Introduction section. (Line 91)
  5. Overall work methodology.is not clear. Pl elaborate it clearly
    • We have elaborated on our methodology by describing each phase in the methodology thoroughly. (Line 377)
  6. Table 1 and 2 should be re-presented. Moreover, all the parameters should be explain clearly .
    • We have changed how Table 1 was presented and explained the purpose of each configuration in Table 2. We have moved Table 2 to the Appendix as we decided it was not crucial to the reader. We also changed the number of Tables 1 and 2 to 3 and A1. (Line 393; Line 717)
  7. Section 4.2 should derive more
    • We tried our best to elaborate on the content of section 4.2 by discussing how these limitations could be solved using image segmentation and object detection. (Line 676)
  8. All Figures should be enhanced at the resolution of 300 dpi.
    • We have replaced all the figures with higher resolution.
  9. Finally, the authors should double-check all formation, typos, and writing throughout the paper.
    • We used Grammarly to double-check all the formatting, typos, and writing.

Reviewer 2 Report

Dear Authors,

In this paper, an image classification model to assess the quality of green space was proposed, trained, evaluated, and implemented. The methods used in the study were selected and implemented in accordance with the purpose.  This research used four evaluation metrics: accuracy, precision, recall, and F1-Score.

More evaluation metrics should be calculated for comparison of the models like Cohen’s kappa, and AUC score.

The recall formula must be corrected as TP/TP+FN.

Figure 1 quality is not suitable.

Author Response

Dear Sir/Madam,

Thank you very much for your time and comments. We have tried to address each comment to the best ability, and we made all the changes mentioned in your comments to the best ability. We have left our response to your kind comments in dark blue font colour below. Thank you for your kind suggestions.

Kind regards,

Jaloliddin Rustamov

  1. More evaluation metrics should be calculated for comparison of the models like Cohen's kappa, and AUC score.
    • Thank you for your kind suggestion. We have added Cohen's Kappa and ROC-AUC as performance metrics in our paper. (Line 542, 545)
  2. The recall formula must be corrected as TP/TP+FN.
    • Thank you for spotting this error. We apologise that we overlooked this error. We have since rectified it. (Line 539)
  3. Figure 1 quality is not suitable.
    • Thank you for your kind suggestion. We have replaced the figure. (Line 386)

Reviewer 3 Report

Overall, it is a well-written paper. Below are my comments:

- I suggest to do not starting sentences/paragraphs with reference numbers. For example, in line 125: '[35] explored various water ....'. There are many of these in the manuscript, please check and rephrase.

- Please add 'Healthy (left), Dried (middle) and Contaminated (right)' to Figure 2's caption.

- It would be good to add a table to show the key differences/similarities between the classification models used in the paper.

-Please sort the bars in Figure 5 from highest to lowest accuracy.

- For Figure 9, there are two things that should be considered. First, the y-axis for the accuracy should start from 0 so it becomes clear the differences in the accuracy are small compared to what is shown in the current figure. Second, you may keep the data labels for file size on the line graph and remove the secondary y-axis for simplicity.

- For the Challenges and Limitations section, how would be possible to address the discussed challenges/limitations?

-In lines 502-503, it was mentioned 'The comparative analysis showed that the proposed model performs better than the related study', so what do you mean?

-Do you think explainable AI could be a measure to be considered as well besides error accuracy, size, and inference time? You may find the following recent surveys helpful:

Arrieta, A. B., Díaz-Rodríguez, N., Del Ser, J., Bennetot, A., Tabik, S., Barbado, A., ... & Herrera, F. (2020). Explainable Artificial Intelligence (XAI): Concepts, taxonomies, opportunities and challenges toward responsible AI. Information fusion, 58, 82-115.

Samek, W., Montavon, G., Lapuschkin, S., Anders, C. J., & Müller, K. R. (2021). Explaining deep neural networks and beyond: A review of methods and applications. Proceedings of the IEEE, 109(3), 247-278.

Saeed, W., & Omlin, C. (2023). Explainable ai (XAI): A systematic meta-survey of current challenges and future opportunities. Knowledge-Based Systems, 110273.

Author Response

Dear Sir/Madam,

Thank you very much for your time and comments. We have tried to address each comment to the best ability, and we made all the changes mentioned in your comments to the best ability. We have left our response to your kind comments in dark blue font colour below. Thank you very much for your kind suggestions.

Kind regards,

Jaloliddin Rustamov

  • I suggest to do not starting sentences/paragraphs with reference numbers. For example, in line 125: '[35] explored various water ....'. There are many of these in the manuscript, please check and rephrase.
    • Thank you for your kind suggestion. We tried our best to change the structure of the sentences or paragraphs to ensure they do not begin with reference numbers.
  • Please add 'Healthy (left), Dried (middle) and Contaminated (right)' to Figure 2's caption.
    • We have replaced Figure 2 and added labels below each image to ensure the reader can understand which image belongs to which class. (Line 406)
  • It would be good to add a table to show the key differences/similarities between the classification models used in the paper. Thank you for your kind suggestion.
    • We added a comparison table of models used in this paper. (Line 525)
  • Please sort the bars in Figure 5 from highest to lowest accuracy.
    • Thank you for your suggestion. We have sorted the bar chart and replaced Figure 5. (Line 591)
  • For Figure 9, there are two things that should be considered. First, the y-axis for the accuracy should start from 0 so it becomes clear the differences in the accuracy are small compared to what is shown in the current figure. Second, you may keep the data labels for file size on the line graph and remove the secondary y-axis for simplicity.
    • We have made a few changes to the graph and replaced it in the paper. We decided to include all the models in the evaluation. (Line 624)
  • For the Challenges and Limitations section, how would be possible to address the discussed challenges/limitations?
    • We did our best to address how the challenges or limitations could be solved. (Line 676)
  • In lines 502-503, it was mentioned 'The comparative analysis showed that the proposed model performs better than the related study', so what do you mean?
    • I would like to ask you to kindly ignore that. We removed it from our paper.
  • Do you think explainable AI could be a measure to be considered as well besides error accuracy, size, and inference time?
    • Thank you for your kind comment. We tried to implement Explainable AI into our paper using Grad-CAM visualisations. (Line 640)

Round 2

Reviewer 1 Report

Thank you for accepting my comments. Manuscript may suitable for publishing in this platform.

Finally, the authors should double-check all formation, typos, and writing throughout the paper.

Author Response

Dear Sir/Madam,

Thank you very much for your time and comment. We have checked the formation, typos, and writing style in the revised manuscript. Thank you for your suggestion once again.

Kind regards,

Jaloliddin Rustamov